# Acute Respiratory Distress Syndrome and Time to Weaning Off the Invasive Mechanical Ventilator among Patients with COVID-19 Pneumonia

**DOI:** 10.3390/jcm10132935

**Published:** 2021-06-30

**Authors:** Jose Bordon, Ozan Akca, Stephen Furmanek, Rodrigo Silva Cavallazzi, Sally Suliman, Amr Aboelnasr, Bettina Sinanova, Julio A. Ramirez

**Affiliations:** 1Washington Health Institute, Washington, DC 20017, USA; bettina_sinanova@yahoo.com; 2Department of Medicine, George Washington University Medical Center, Washington, DC 20037, USA; 3Center of Excellence for Research in Infectious Diseases (CERID), University of Louisville, Louisville, KY 40292, USA; ozan.akca@louisville.edu (O.A.); stephen.furmanek@louisville.edu (S.F.); rodrigo.cavallazzi@louisville.edu (R.S.C.); amr.aboelnasr@louisville.edu (A.A.); j.ramirez@louisville.edu (J.A.R.); 4Department of Anesthesiology and Perioperative Medicine, University of Louisville, Louisville, KY 40292, USA; 5Comprehensive Stroke Clinical Research Program (CSCRP), University of Louisville, Louisville, KY 40292, USA; 6Division of Infectious Diseases, University of Louisville, Louisville, KY 40292, USA; 7Division of Critical Care Medicine, University of Louisville, Louisville, KY 40292, USA; sally.suliman@louisville.edu

**Keywords:** COVID-19, SARS-CoV-2, CAP, pneumonia, ARDS, IMV

## Abstract

Acute respiratory distress syndrome (ARDS) due to coronavirus disease 2019 (COVID-19) pneumonia is the main cause of the pandemic’s death toll. The assessment of ARDS and time on invasive mechanical ventilation (IMV) could enhance the characterization of outcomes and management of this condition. This is a city-wide retrospective study of hospitalized patients with COVID-19 pneumonia from 5 March 2020 to 30 June 2020. Patients with critical illness were compared with those with non-critical illness. We examined the severity of ARDS and other factors associated with (i) weaning patients off IMV and (ii) mortality in a city-wide study in Louisville, KY. Of 522 patients with COVID-19 pneumonia, 219 (41.9%) were critically ill. Among critically ill patients, the median age was 60 years; 53% were male, 55% were White and 32% were African American. Of all critically ill patients, 52% had ARDS, and 38% of these had severe ARDS. Of the 25% of patients who were weaned off IMV, those with severe ARDS were weaned within eleven days versus five days for those without severe ARDS, *p* = 0.023. The overall mortality for critically ill patients was 22% versus 1% for those not critically ill. Furthermore, the 14-day mortality was 31% for patients with severe ARDS and 12% for patients without severe ARDS, *p* = 0.019. Patients with severe ARDS versus non-severe ARDS needed twice as long to wean off IMV (eleven versus five days) and had double the 14-day mortality of patients without severe ARDS.

## 1. Introduction

As of 22 June 2021, there have been 3,876,081 losses of human life reported worldwide due to the COVID-19 pandemic [1]. COVID-19 pneumonia resulting in critical illness is the main cause of the high death toll of this pandemic. Up to 39.7% of patients with COVID-19 have been reported to develop critical illness—primarily associated with acute respiratory distress syndrome (ARDS)—in a predominantly African American cohort [2]. Furthermore, the mortality rate of critically ill patients with COVID-19 has been reported at up to 98% [3]. The characteristics and major outcomes of critically ill patients due to COVID-19 pneumonia on a US city-wide scale have not yet been reported. ARDS is a major outcome of COVID-19 patients that requires a remarkably long invasive mechanical ventilation (IMV). The time to wean patients off IMV is a turning point in prognosis, at which point aggressive clinical management can be de-escalated.

Indeed, it is critically important to determine the associated factors and the average time to wean COVID-19 patients off of IMV. An optimal characterization of critically ill patients with COVID-19 from a population representative of the US as a whole could provide balanced data for the development of guidelines regarding the appropriate assessment and management of COVID-19. The city of Louisville, KY, has been recognized as a population with sociodemographic, economic, and health-related characteristics that are representative of the US [4]. The 2019 population of the city of Louisville was 766,557, of whom 16.7 % were persons 65 years and older, 51.7% were women, 71.8% were Caucasians, and 22.4% were African Americans; moreover, median house income was $54,357 and 15.4% of the population was living in poverty [5]. 

We measured the outcomes of patients who were critically ill due to COVID-19 pneumonia; we examined the severity of ARDS and factors associated with (i) weaning patients off IMV and (ii) mortality in a city-wide study in Louisville, KY, USA. 

## 2. Methods

### 2.1. Study Design, Subjects, and Setting

This study was a secondary data analysis of the Louisville Coronavirus Surveillance Program, KY, USA, a city-wide retrospective observational study of consecutive unduplicated adult patients with COVID-19 pneumonia who required admission to any of the city’s 8 hospitals between 5 March 2020—when the first hospitalized patient with COVID-19 was identified in Louisville—and 1 July 2020 [6]. Our study group comprised patients with COVID-19 pneumonia who were critically ill. The control group consisted of patients with COVID-19 pneumonia who were not critically ill and were admitted to any of the eight hospitals in the city of Louisville, KY, USA. Patient follow-up ceased at hospital discharge. We examined the severity of ARDS and other factors associated with (i) weaning patients off IMV, and (ii) mortality in a city-wide study in Louisville, KY. Appendix A.

### 2.2. Severity of Illness Definitions

The severity of illness was defined during the first 24 h of hospitalization in accordance with National Institutes of Health (NIH) guidelines for the management of COVID-19 [7]. Only patients with Stages 3 to 9 COVID-19 were included in this study, and Stages 6 to 9 were considered critical illness. In the cases of patients with critical illness, the analysis used data recorded from the point at which they fulfilled the criteria of having critical illness, either at the time of hospitalization or thereafter.

### 2.3. Mortality Prediction

COVID-19 pneumonia at the time of hospitalization and mortality prediction were evaluated using two well-established pneumonia scores, the pneumonia severity index (PSI) score [8] and the CURB-65 score [9].

### 2.4. Exclusion Criteria

Patients without COVID-19 and patients who did not require hospital admission were excluded from this study. Patients who had do-not-resuscitate (DNR) or do-not-intubate (DNI) orders upon admission and individuals younger than 18 years were also excluded.

### 2.5. Predictor Variables

Predictor variables included clinical characteristics, laboratory findings, and imaging results at the time of hospitalization. In cases of critical SARS-CoV-2 CAP, these variables were collected within 24 h of the time at which the patients fulfilled the definition of critical COVID-19, either at hospital admission or during hospitalization.

### 2.6. Outcome Variables

i. Time to wean off IMV was truncated at 21 days. Weaning success was eliminated if the patient died soon thereafter. Patients who remained on IMV, or were weaned off but expired while in the hospital, were considered instances of failure to wean off IMV. In the event that a patient weaned off IMV died in the hospital, the time to wean off invasive mechanical ventilation was censored and set to 21 days. Patients who underwent tracheostomy were not considered weaned until after decannulation.

ii. Time to mortality was truncated at 30 days. Patients who were discharged alive before 30 days of hospitalization were not followed after discharge and were given a right-censored time to mortality consisting of the duration at their date of hospital discharge.

### 2.7. Statistical Analysis

Patient characteristics were reported as frequencies and percentages or as medians and interquartile ranges, for categorical and continuous data, respectively. Differences in baseline patient characteristics were assessed by chi-squared tests of independence or Mann–Whitney U tests for categorical and continuous data, respectively.

Inflammatory laboratory values of lactate, ferritin, interleukin-6 (IL-6), D-dimer, and c-reactive protein (CRP) were dichotomized into high/low values based on receiver operating curve (ROC) values that maximized sensitivity and specificity to predict in-hospital mortality. Time-to-event data were represented by Kaplan–Meier curves split by severity of ARDS. Log-rank tests were performed for unadjusted comparisons. To determine protective and risk factors for time to event data, multivariable, stratified Cox proportional hazards regressions with cluster-robust standard errors were performed. Variables were selected through forward and backward stepwise selection, based on minimizing the Akaike information criterion (AIC). The variables used in the selection process were (i) patient demographics of age, sex, race, and ethnicity; (ii) history of the following variables: neoplastic disease within the past year, renal disease, heart failure, chronic obstructive pulmonary disease, smoking history, diabetes, obesity, hypertension, hyperlipidemia; (iii) standardized values of temperature, respiratory rate, mean arterial pressure, heart rate, blood urea nitrogen (BUN), serum glucose, serum sodium, and hematocrit; (iv) dichotomized laboratory values of IL-6, CRP, d-dimer, ferritin, and lactate; and (v) exam findings of altered mental status and computed tomography (CT) or chest x-ray findings of pleural effusion and severe ARDS. Any *p*-value less than 0.05 was considered statistically significant. All analysis was performed in R version 3.5.1. (Vienna, Austria.)

## 3. Results

### 3.1. Study Population and Main Characteristics

Of the 712 patients hospitalized due to COVID-19 from 5 March 2020 to 30 June 2020 in the city of Louisville, KY, our study included a total of 522 patients with COVID-19 pneumonia (Figure 1). Due to DNR and DNI advanced directives, 67 critically ill patients and 43 non–critically ill patients were excluded. Our study included a final total of 219 critically ill and 303 non–critically ill COVID-19 pneumonia patients. The main characteristics of these patients are shown in Table 1.

On average, critically ill COVID-19 patients were two years older than controls. The population of critically ill male patients was 10% greater than non–critically ill male patients (53% versus. 43% *p* = 0.02). Among critically ill pneumonia patients, African Americans accounted for 32% and Caucasians 55%. Arterial hypertension was present in 58% of critically ill versus 54% of non–critically ill patients, and diabetes mellitus was observed in 40% versus 29%, respectively (Table 1).

### 3.2. Severity of Illness and Outcomes

Overall, critically ill patients with COVID-19 had greater severity of illness than controls, with the most notable difference being the degree of hypoxemia (Table 1); these differences were determined to be statistically significant. The unadjusted outcomes are shown in Table 2. ARDS was present in 52% and shock in 27% of critically ill patients. Cardiovascular events were reported more than three times more often among critically ill patients than those not critically ill: 27% versus 8% respectively.

### 3.3. Weaning Off Invasive Mechanical Ventilation

Critically ill patients exhibited a median arterial partial pressure of oxygen/fraction of inspired oxygen (PaO_2_/FiO_2_) ratio of 174, which is consistent with the oxygenation level of moderate ARDS. Precisely 38% of these patients had severe ARDS. The univariate analysis of risk factors associated with the weaning off the IMV is shown in Appendix A. IL-6 > 65 pg/mL and male sex were associated with the lowest statistically significant likelihood of weaning off IMV (Figure 2). Twenty-five percent of patients with severe ARDS weaned off IMV within eleven days versus five days for those without severe ARDS (Figure 3).

### 3.4. Mortality

The overall mortality was 20% versus 1% for critically and non–critically ill patients, respectively. Among those who expired, critically ill patients were 17 years younger than those who were non–critically ill, and the male to female ratio was 2.68. The univariate analysis of risk factors associated with hospital mortality is shown in Appendix A. After stepwise selection for time to hospital mortality and adjustment by age, African American patients had a 53% lower risk of death (Figure 4). Additionally, arterial hypertension, severe ARDS, and an increase in serum sodium were significantly associated with risk of hospital mortality. Mortality difference between patients with and without severe ARDS was minimal within the first seven days but became substantially wider thereafter. By Day 14, 12% of patients without severe ARDS expired compared with 31% of patients with severe ARDS, *p* = 0.009 (Figure 5).

## 4. Discussion

### 4.1. Patients Characteristics

Our study demonstrated that 41.9% of patients with COVID-19 pneumonia developed critical illness, which was largely associated with ARDS. The main characteristics of our critically ill patients included a median age of 60 years, male sex (53%), Black (32%), White (55%), current or former tobacco smoker (37%), and diabetes mellitus (36%). Therefore, a 60-year-old male who is a former or current smoker suffering from diabetes mellitus seems to be at high risk of becoming critically ill with COVID-19 pneumonia. Considering the racial distribution of the city of Louisville by the 2019 US census, African Americans were overrepresented by 10% and Whites were underrepresented by 16% among those suffering from critical COVID-19 pneumonia.

In relation to the sex distribution for critical illness, females were 7% more prevalent among those with non-critical illness and 3% less among those with critical illness. The 10% increase in male sex from non-critical to critical COVID-19, and vice versa for female sex, demands further research. A lower percentage of females in relation to the severity of illness has been similarly reported by others [3,10,11,12]. This raises the binary question: are men at greater risk for critical COVID-19 pneumonia, or are women better protected against it? Female protection against critical COVID-19 pneumonia has been linked to the estrogen immunomodulatory and anti-inflammatory activity of high physiological concentrations of the steroids 17β-estradiol (E2) and progesterone (P4). It has been postulated that E2 and P4 decrease innate immune inflammatory responses and increase immune tolerance and antibody production. Thus, E2 and P4 may lessen the immune dysregulation that leads to the COVID-19 cytokine storm [13]. Many research studies on the effects of estrogen and COVID-19 are currently underway [14].

### 4.2. ARDS and the Time to Wean Off IMV of Patients with Critical COVID-19 Pneumonia

Our study showed that over half (52%) of critically ill patients had ARDS, and almost one third (28%) had severe ARDS. Furthermore, among patients who were weaned off IMV (25% of the study cohort), those with severe ARDS required IMV for a longer period than those with non-severe ARDS (15 versus 8 days, *p* = 0.016). Since the pre-COVID era, late weaning from IMV has been associated with poor subsequent quality of life [15]. The severity of ARDS and length of IMV-dependent time exhibited by patients in our study bear some similarity to non–COVID-19 pneumonia ARDS, including influenza ARDS. The Lung Safe Study reported ARDS in 59.4% of patients with pneumonia, severe ARDS in 23.43%, and a median IMV-dependence period of 8 days [16]. Limited data on influenza pneumonia and ARDS are available in the literature. In a comparison of critical COVID-19 pneumonia and influenza in a cohort of 201 adults, Hernu et al. reported a mean of IMV-dependence of 14 days; 56% of the patients had ARDS and 75% of those with ARDS expired [17]. Martinez et al. studied 595 patients with influenza who were admitted to the ICU; the mean age was 59 years, 37.8% were male, pneumonia was present in 34.5%, and the mean period of IMV-dependence was 14 days. In this study, ARDS was present in 51.9% of patients and mortality was 20.3% among those with ARDS [18]. The higher rate of ARDS in patients with COVID-19 and the long period of IMV-dependence among our patients with severe ARDS are likely due to the higher prevalence of comorbidities associated with chronic inflammation and the not fully identified mechanisms of exuberating lung inflammation [19,20]. In our study, IL-6 and male sex were associated with failing to wean off IMV. IL-6 has been reported to be an independent and significant predictor of disease severity and death [21]. IL-6 was related to ACE-2 activity, which has been directly associated with the cytokine storm, and ACE-2 polymorphism could explain the variability of the rate of ARDS among patients and its lower incidence among female sex [22].

### 4.3. Mortality among COVID-19 Pneumonia

The mortality of patients in our study with critical illness was about 20 times that of patients without critical illness: 20% versus 1% (see Table 2). Furthermore, mortality was more notable and statistically significant after Day 7. The cytokine storm and, in particular, the high level of IL-6 have been strongly correlated with mortality [23]. The greater mortality of critical COVID-19 pneumonia is very likely related to the severity of illness among people suffering from comorbidities associated with chronic inflammation and ongoing organ damage [24]. Our patients with critical COVID-19 presented with more than three times as many cardiac events as those without critical COVID-19. The association of male sex, high IL-6, and ARDS could explain the higher mortality of patients with critical COVID-19.

The greatest strength of our study is its characterization of the critical illness and outcomes of patients with COVID-19 pneumonia across an entire city that is demographically representative of the US population. The large number of patients included in our study is an additional strength. The limitations of the study include the retrospective study design and missing data concerning some of the patients’ characteristics.

In summary, more than two fifths (41.9%) of patients admitted with COVID-19 pneumonia became critically ill and more than half (52%) of them developed ARDS. Among the 25% of patients who were weaned off IMV, those with severe ARDS required about two weeks on IMV, compared to about one week for those with non-severe ARDS. The two-week mortality of patients with severe ARDS (31%) was more than double that of patients without severe ARDS (12%). Future research should focus on signaling mechanisms leading to deleterious inflammation and ARDS resulting in a high morbidity and mortality of patients with COVID-19 pneumonia.

## Figures and Tables

**Figure 1 jcm-10-02935-f001:**
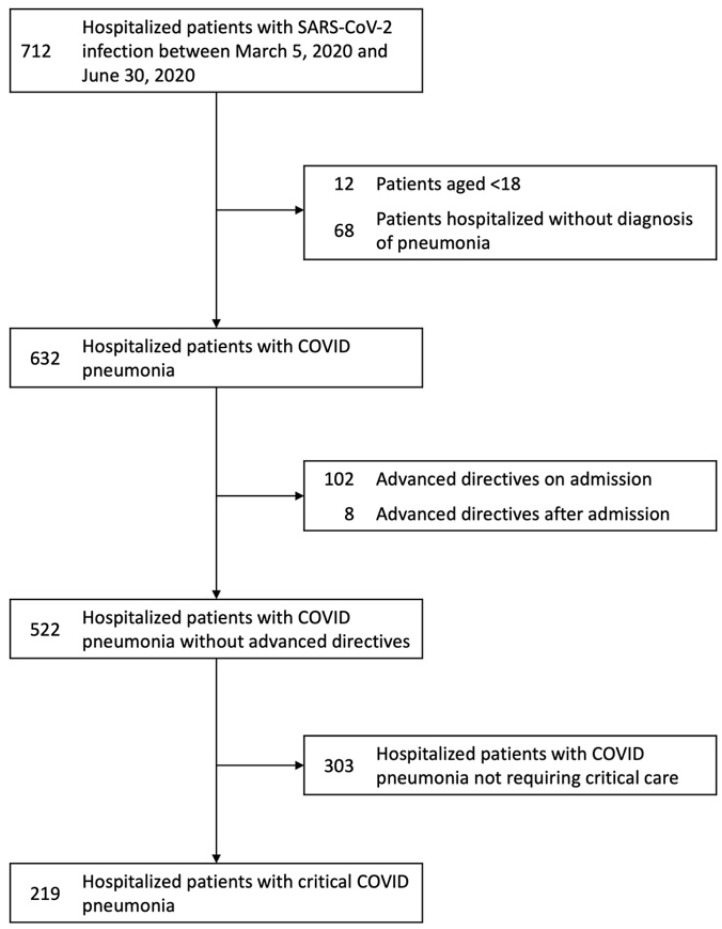
Flowchart of patients with COVID-19 hospitalized in the city of Louisville, KY, from 5 March 2020 to 30 June 2020. The 219 patients with critical COVID-19 were selected upon elimination of 67 patients with do-not-resuscitate and do-not-intubate advanced directive orders. Similarly, the 303 patients with non-critical COVID-19 pneumonia resulted in the elimination of 43 patients. Abbreviations: COVID-19, coronavirus disease 2019; DNI, do-not-intubate; DNR, do-not-resuscitate; SARS-CoV-2, severe acute respiratory syndrome coronavirus-2.

**Figure 2 jcm-10-02935-f002:**
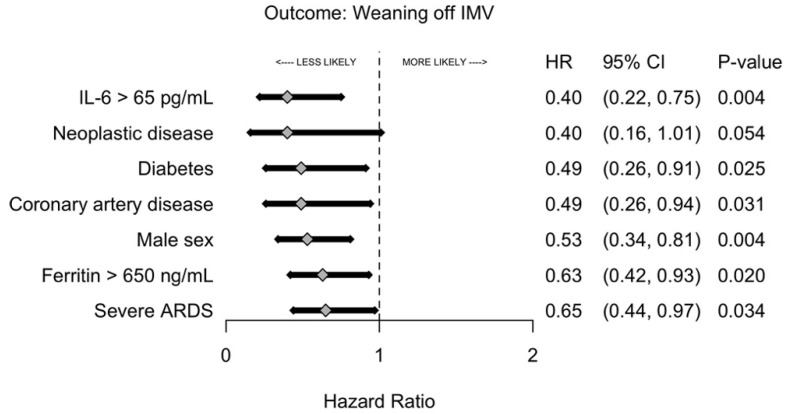
Likelihood of weaning off invasive mechanical ventilation by risk factors. The forest plot analysis is a multivariable logistic regression with cluster-robust standard errors that was selected using a forward and backward stepwise selection procedure based on overall model likelihood. Abbreviations: ARDS, acute respiratory distress syndrome; IL-6, interleukin-6; IMV, invasive mechanical ventilation.

**Figure 3 jcm-10-02935-f003:**
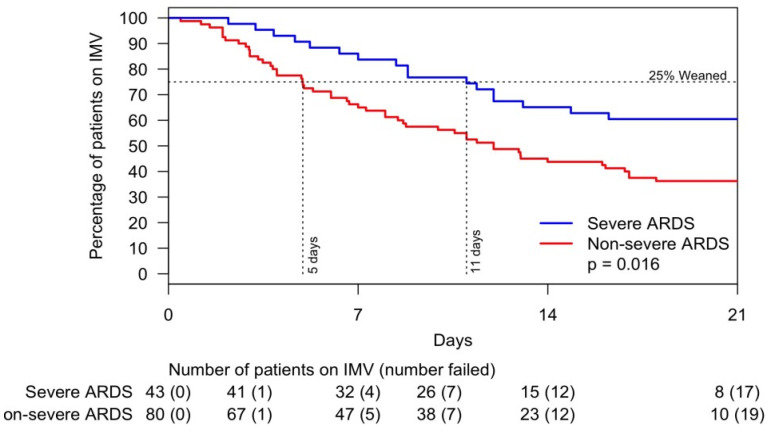
Likelihood of weaning off invasive mechanical ventilation within 21 days by severe and non-severe ARDS. Abbreviations: ARDS, acute respiratory distress syndrome; IMV, invasive mechanical ventilation.

**Figure 4 jcm-10-02935-f004:**
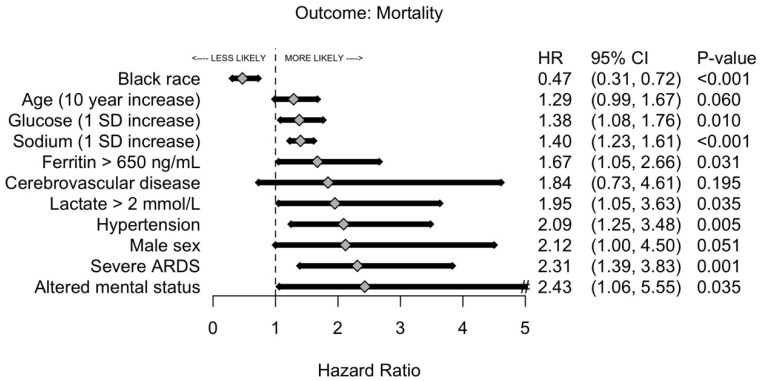
Likelihood of mortality of patients with critical COVID-19 pneumonia by risk factors. The forest plot analysis is a multivariable logistic regression with cluster-robust standard errors that was selected using a forward and backward stepwise selection procedure based on overall model likelihood. Abbreviations: ARDS, acute respiratory distress syndrome; COVID-19, coronavirus disease 2019.

**Figure 5 jcm-10-02935-f005:**
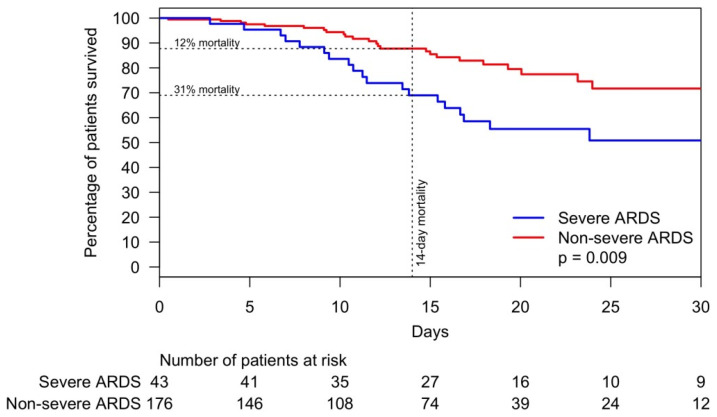
Time to mortality of patients with COVID-19 pneumonia within 28 days by severe and non-severe ARDS. Abbreviations: ARDS, acute respiratory distress syndrome; COVID-19, coronavirus disease 2019.

**Table 1 jcm-10-02935-t001:** Characteristics of patients with COVID-19 pneumonia at the time of assignment to the groups of critical and non-critical illness.

	Critical Illness	Non-Critical Illness	*p* Value
	(*n* = 219)	(*n* = 303)	
Demographics			
Age (median (IQR))	60 (50, 70)	58 (44, 71)	0.234
Male sex (%)	116 (53)	129 (43)	0.024
Race (%)			0.170
White	121 (55)	144 (48)	
Black	69 (32)	105 (35)	
Other	29 (13)	54 (18)	
Hispanic ethnicity (%)	21 (10)	50 (17)	0.032
Neoplastic disease (%)	14 (6)	14 (5)	0.490
Renal disease (%)	34 (16)	49 (16)	0.938
Liver disease (%)	4 (2)	13 (4)	0.188
Heart failure (%)	28 (13)	29 (10)	0.308
COPD (%)	33 (15)	33 (11)	0.199
Smoking history, current or former (%)	82 (37)	100 (33)	0.338
Diabetes (%)	79 (36)	84 (28)	0.053
Obese (%)	121 (55)	154 (51)	0.362
Hypertension (%)	116 (53)	155 (51)	0.749
Coronary artery disease (%)	24 (11)	40 (13)	0.525
Hyperlipidemia (%)	72 (33)	97 (32)	0.910
Exam and lab values at time of admission or critical illness, median (IQR)			
Temperature (degrees celsius)	37.6 (36.9, 38.8)	37.8 (37.0, 38.5)	0.880
Respiratory rate (breaths/min)	27.0 (21.0, 34.0)	20.0 (18.0, 24.0)	<0.001
Heart rate (beats/min)	101.0 (87.5, 112.5)	97.0 (80.0, 110.0)	0.050
Systolic blood pressure (mmHg)	110.0 (97.0, 132.0)	124.0 (109.0, 141.0)	<0.001
Diastolic blood pressure (mmHg)	61.0 (48.0, 75.0)	69.0 (61.0, 81.5)	<0.001
Mean arterial pressure (mmHg)	77.7 (64.7, 92.3)	88.7 (78.3, 99.0)	<0.001
BUN (mg/dL)	17.0 (12.0, 29.0)	15.0 (10.0, 23.0)	0.021
Glucose (mg/dL)	132.0 (112.0, 183.0)	117.0 (102.0, 143.0)	<0.001
Hematocrit (%)	37.4 (33.0, 40.7)	39.0 (35.9, 42.6)	<0.001
Sodium (mEq/L)	136.0 (133.0, 138.0)	136.0 (134.0, 138.0)	0.544
White blood cell count (×1000 per uL)	7.8 (5.8, 11.0)	5.6 (4.3, 7.3)	<0.001
Neutrophil count (×1000 per uL)	5.8 (4.5, 8.3)	3.7 (2.7, 5.6)	<0.001
Lymphocyte count (×1000 per uL)	0.9 (0.7, 1.1)	1.1 (0.8, 1.4)	<0.001
Neutrophil/lymphocyte (ratio)	8.0 (5.2, 10.1)	3.4 (2.2, 5.9)	<0.001
Interleukin-6 (pg/mL)	92.2 (51.3, 218.1)	33.8 (16.6, 58.2)	<0.001
C-reactive protein (CRP) (mg/L)	162.4 (82.2, 216.4)	53.0 (24.5, 120.0)	<0.001
D-dimer (ng/mL)	813.0 (318.2, 2147.5)	681.0 (370.0, 1080.0)	0.115
Ferritin (ng/mL)	554.5 (289.8, 994.8)	292.0 (115.5, 637.0)	<0.001
Lactate (mmol/L)	1.5 (1.2, 2.4)	1.2 (1.0, 1.7)	<0.001
Creatinine (mg/dL)	1.0 (0.7, 1.4)	0.9 (0.7, 1.2)	0.024
Albumin (g/dL)	3.3 (2.8, 3.6)	3.8 (3.5, 4.1)	<0.001
Bilirubin (mg/dL)	0.7 (0.5, 1.0)	0.6 (0.4, 0.9)	<0.001
PaO_2_/FiO_2_ (ratio)	174.0 (85.4, 253.0)	355.0 (336.8, 378.5)	<0.001
SpO_2_/FiO_2_ (ratio)	220.0 (95.8, 332.1)	442.9 (384.5, 457.1)	<0.001
Treatments and therapies, *n* (%)			
Hydroxychloroquine (%)	120 (55)	76 (25)	<0.001
Azithromycin (%)	158 (72)	181 (60)	0.001
Remdesivir (%)	21 (10)	7 (2)	<0.001
Steroids (%)	91 (42)	28 (9)	<0.001
Low molecular weight heparins (%)	150 (68)	180 (59)	0.092
Heparin (%)	69 (32)	17 (6)	<0.001
Plasma (%)	36 (16)	7 (2)	<0.001

Patients with advanced directives were not included. Abbreviations: BUN, blood urea nitrogen; COPD, chronic obstructive pulmonary disease; COVID-19, coronavirus disease 2019; IQR, interquartile range; PaO_2_/FiO_2_, arterial partial pressure of oxygen/fraction of inspired oxygen; SpO_2_/FiO_2_, peripheral blood oxygen saturation/fraction of inspired oxygen.

**Table 2 jcm-10-02935-t002:** Clinical outcomes among all patients hospitalized with COVID-19 pneumonia.

	Critical Illness	Non-Critical Illness	*p* Value
	(*n* = 219)	(*n* = 303)	
Outcomes			
Time to death or discharge from admission	14.0 (8.0, 20.0)	4.1 (2.6, 6.9)	<0.001
Time to ICU admission from admission	1.0 (0.0, 2.5)		
Time to death or ICU discharge from ICU admission	8.3 (2.7, 15.2)		
Time to IMV from admission	1.7 (0.3, 3.7)		
Time to death, tracheostomy, or weaning from intubation	8.8 (4.0, 14.2)		
Time to IMV from ICU admission	0.1 (0.0, 1.4)		
Septic shock (%)	60 (27)	0 (0)	<0.001
DIC (%)	2 (1)	0 (0)	0.343
Development of ARDS (%)	114 (52)	0 (0)	<0.001
Cardiovascular or cardiac event (%)	59 (27)	24 (8)	<0.001
In-hospital mortality (%)	43 (20)	4 (1)	<0.001
Age	66 (57, 73)	83 (75, 90)	<0.001
Male sex	29 (67)	1 (25)	0.124

Patients with advanced directives were not included; Abbreviations: ARDS, acute respiratory distress syndrome; COVID-19, coronavirus disease 2019; DIC, disseminated intravascular coagulation; ICU, intensive care unit; IMV, invasive mechanical ventilation.

## Data Availability

Summary data are included in the Appendix A section of the online article. Other data are available on request.

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
