# Peer review of "Acute Respiratory Distress Syndrome and Time to Weaning Off the Invasive Mechanical Ventilator among Patients with COVID-19 Pneumonia"

_jcm, 2021, doi:10.3390/jcm10132935_

Round 1

Reviewer 1 Report

Dear Dr. Jose Bordon,

Thank you for sending this intriguing manuscript for review.
I thoroughly enjoyed the read. However, I have a few comments, which might help you improve the quality of your manuscript further. 

Title: I relish eye-catching titles. However, in my opinion, general relatability in such a title is of utmost importance. Dr. Osler's work should be common knowledge; almost all millennials and, indeed, all of the gen Z cannot relate to the captain of the death with wide-ranging guesses from HIV to Ebola and malignancy to COVID-19.  Hence, due to poor relatability, I would recommend removing it from the title.

Abstract: It summarizes the study well.
Introduction: The introduction is well-written. I have two suggestions.
1. Update the figure of deaths caused by COVID-19 as it is reflecting the deaths from 6 months back.
2. Page 2, lines 41-47: Beautiful piece of writing. However, as the article does not delve into the technological, global epidemiologic, or economic effects of the pandemic, the relevance of these lines is questionable in the context of the article. 

Methods: Page 3, line 97: predictor variables are mentioned under the exclusion criteria. The description of predictor variables is better suited as either under a separate heading or moving them before the inclusion criteria section.
Page 3, lines 108-109:  Regarding censoring, the assumption should be that the censored patients should have the same survival prospects as those who continue to be followed up (non-informative censoring.) Patients discharged
alive should not have been censored; authors should consider their status
as event-free. 
Also, the hypothesis is not stated clearly. It may be that I am not able to read between the lines, but extrapolating from the results of severe disease leading to more mortality, the null hypothesis here would be "increase in disease severity in COVID-19 is not related to mortality." This would be counter-intuitive and is an established fact in ARDS in general, and COVID-19 associated respiratory disease of varying severity in particular.

Suggestion regarding methodology robustness: I would like to recommend using a propensity-matched historical cohort from one of the prior respiratory epidemics, e.g., Influenza or MERS or SARS-CoV-1 to work out the factors/variables associated with mortality associated with COVID-19 ARDS.

Results and Discussion: Results are appropriately presented, and discussion is relevant.

Ethical approval: Ethical approval from an institutional body is not explicitly stated. Stating it would be relevant to the manuscript.

I wish you the best of luck with this manuscript and any future endeavors.

Author Response

We greatly appreciate to the reviewer for such thoughtful corrections and suggestions according to the reviewer’s comments text orders. Our reply is in bold fonts.  

1.Title: I relish eye-catching titles. However, in my opinion, general relatability in such a title is of utmost importance. Dr. Osler's work should be common knowledge; almost all millennials and, indeed, all of the gen Z cannot relate to the captain of the death with wide-ranging guesses from HIV to Ebola and malignancy to COVID-19.  Hence, due to poor relatability, I would recommend removing it from the title. 

We agree with the reviewer. We remove the phrase “The return of the captain of Death.” 

  1. Abstract: It summarizes the study well. Introduction: The introduction is well-written. I have two suggestions. Update the figure of deaths caused by COVID-19 as it is reflecting the deaths from 6 months back. 

The study time was from March 5th, 2020 to July 1th 2020. The study design only includes in hospital death. We do not have the data for the 6 months back and we’re not sorry not being able to provide the deaths from 6 months back.  

  1. Page 2, lines 41-47: Beautiful piece of writing. However, as the article does not delve into the technological, global epidemiologic, or economic effects of the pandemic, the relevance of these lines is questionable in the context of the article.  

We agree with the reviewer. We remove the texts from line 41-47. 

  1. Methods: Page 3, line 97: predictor variables are mentioned under the exclusion criteria. The description of predictor variables is better suited as either under a separate heading or moving them before the inclusion criteria section. 

We agree with the reviewer. We created the heading 3.5. Predictor variables corresponding to line 97. Consequently, the numbers for the two following headings were corrected.   

  1. Page 3, lines 108-109:  Regarding censoring, the assumption should be that the censored patients should have the same survival prospects as those who continue to be followed up (non-informative censoring.) Patients discharged alive should not have been censored; authors should consider their status as event-free.  Manuscript text 108-109: Time to mortality during hospitalization was truncated at 30 days. Patients who were discharged alive were censored at their date of hospital discharge.

                             We agree with the reviewer that this language was unclear. Given that the event is mortality and that patients were not followed after hospitalization, we agree that patients who are discharged alive should be considered as event-free. Any patient that was discharged before the time of truncation (30 days) would be considered lost to follow-up and event-free, and would thus be right-censored, as right censoring occurs when the time to true event time (mortality) is greater than the time in the study. We have revised this sentence to state the following: “Time to mortality was truncated at 30 days. Patients who were discharged alive before 30 days were not followed after discharge and were given a right-censored time to mortality consisting of their length of stay.” 

  1. Also, the hypothesis is not stated clearly. It may be that I am not able to read between the lines, but extrapolating from the results of severe disease leading to more mortality, the null hypothesis here would be "increase in disease severity in COVID-19 is not related to mortality." This would be counter-intuitive and is an established fact in ARDS in general, and COVID-19 associated respiratory disease of varying severity in particular.

Our study objective measures the outcomes of severity of ARDS and factors associated in relation to time to wean off IMV and mortality. In this regard, we use as reference cases without severe ARDS. We did not claim nor intend to prove that the outcome of severe ARDS is worse than non-severe ARDS. A qualitative perspective that a severe is worse ARDS than non-severe ARDS is not enough. We believe that the numbers of our study outcomes could guide physicians in building up appropriate expectations in relation to severity of ARDS and time to wean off IMV and mortality. We corrected the text accordingly indicated below. 

Original objective crossed text: In an effort to characterize the outcomes of patients who were

critically ill due to COVID-19 pneumonia, we examined the severity of ARDS and factors associated

with (i)  weaning patients off IMV and (ii) mortality in a city-wide study in Louisville, KY.  

Corrected objective underlined text: We measure the outcomes of patients who were critically ill

Due to COVID-19 pneumonia, we examined the severity of ARDS and factors associated

with (i) weaning  patients off IMV and (ii) mortality in a city-wide study in Louisville, KY.  

  1. Suggestion regarding methodology robustness: I would like to recommend using a propensity-matched historical cohort from one of the prior respiratory epidemics, e.g., Influenza or MERS or SARS-CoV-1 to work out the factors/variables associated with mortality associated with COVID-19 ARDS. 

We agree with the reviewer that a propensity-matched historical cohort from one of the prior respiratory epidemics to work out the factors/variables associated with mortality associated with COVID-19 ARDS. Unfortunately, we do not have such cohort data. Thank you. 

Reviewer 2 Report

Overall, the authors must be commended for an excellent manuscript on Severe COVID-19 ARDS, their characteristics, mortality and time to wean of invasive mechanical ventilation. The study appears of sound methodology and results are interesting. I have only few comments for the authors:

  1. Title can be shortened as it is extensively long.
  2. There are some self-citations in the manuscript.
  3. I would like to clarify if the authors studied the same cohort as in citation 7 as some of the findings are similar.
  4. The title and the description of results is rather misleading as the study is so thorough and captures so many risk factors associated with poor outcome in severe COVID-19 ARDS patients but the authors chose to title it on time to weaning off IMV which is rather a secondary outcome. It would be more interesting to consolidate the manuscript and only highlight risk factors associated with mortality in severe COVID-19 ARDS and mention time to weaning off IMV as a secondary result.
  5. Mortality has to be specified as 30 day mortality as that was where it was truncated by the authors. We already know that some of the critically ill patients can have long and protracted clinical course that can extend beyond 30 days.

Author Response

We greatly appreciate the reviewer for such thoughtful corrections and suggestions according to the reviewer’s comments text orders. Our reply is in bold fonts.  

 Overall, the authors must be commended for an excellent manuscript on Severe COVID-19 ARDS, their characteristics, mortality and time to wean of invasive mechanical ventilation. The study appears of sound methodology and results are interesting. I have only few comments for the authors: 

  1. Title can be shortened as it is extensively long. 

We agree with the reviewer. We shorten the title. 

  1. There are some self-citations in the manuscript. 

We agree about our self-citations and we believe that these citations are pertinent except that the reviewer indicates to remove the citation due to specific reasons. Thank you.   

  1. I would like to clarify if the authors studied the same cohort as in citation 7 as some of the findings are similar. 

Thank you for bringing this specific and important question. It’s very important to indicate that our manuscript study design and consequently outcomes are different from the citation 7. Our study focused on the severity of ARDS, time to wean off IMV and mortality from a population who was critically ill. The citation 7 study focused on all patients admitted in the hospital, did not differentiate between severe vs non-severe ARDS, time to wean off IMV and mortality in relation to ARDS. Thank you.   

  1. The title and the description of results is rather misleading as the study is so thorough and captures so many risk factors associated with poor outcome in severe COVID-19 ARDS patients but the authors chose to title it on time to weaning off IMV which is rather a secondary outcome. It would be more interesting to consolidate the manuscript and only highlight risk factors associated with mortality in severe COVID-19 ARDS and mention time to weaning off IMV as a secondary result. 

We agree with the reviewer that the title could misguide given the large numbers of variables measures. There two points that we believe assist the readers to have the optimal understanding of the title and results.  

The methods section provides a good description of the risk factors evaluated in the study. Therefore the methods section provide good guidance of risk factor measurements. 

We were told to optimize the study objectives and we did as it read below. We believe that study objectives remediate any potential misleading of the study title.  

             Original objective crossed text: In an effort to characterize the outcomes of patients who  

             were critically ill due to COVID-19 pneumonia, we examined the severity of ARDS and factors 

              associated with (i) weaning patients off IMV and (ii) mortality in a city-wide study in  

              Louisville, KY.  

            Corrected objective underlined text: We measure the outcomes of patients who were  

            critically ill due to COVID-19 pneumonia, we examined the severity of ARDS and factors 

            associated with (i) weaning patients off IMV and (ii) mortality in a city-wide study in  

            Louisville, KY.  

  1. Mortality has to be specified as 30 day mortality as that was where it was truncated by the authors. We already know that some of the critically ill patients can have long and protracted clinical course that can extend beyond 30 days.  

We agree that outcomes after 30 days not being assessed are a limitation of our manuscript, and have added the following to the discussion: “As critically-ill patients may have prolonged clinical course, another limitation of the study was that mortality after 30 days was not assessed.” 

Round 2

Reviewer 2 Report

I am satisfied by the authors' response and the changes in the manuscript and would like to recommend it for publication.